# Juvenile Autoimmune Hepatitis: Recent Advances in Diagnosis, Management and Long-Term Outcome

**DOI:** 10.3390/diagnostics13172753

**Published:** 2023-08-24

**Authors:** Silvia Nastasio, Antonella Mosca, Tommaso Alterio, Marco Sciveres, Giuseppe Maggiore

**Affiliations:** 1Division of Gastroenterology, Hepatology & Nutrition, Boston Children’s Hospital and Harvard Medical School, Boston, MA 02115, USA; silvia.nastasio@childrens.harvard.edu; 2Hepatogastroenterology, Rehabilitative Nutrition, Digestive Endoscopy and Liver Transplant Unit, ERN RARE LIVER, Bambino Gesù Children’s Hospital, IRCCS, 00165 Rome, Italy; antonella.mosca@opbg.net (A.M.); tommaso.alterio@opbg.net (T.A.); 3Pediatric Department and Transplantation, ISMETT, 90133 Palermo, Italy; msciveres@ismett.edu

**Keywords:** autoimmune hepatitis, acute liver failure, autoimmune liver disease, active chronic hepatitis liver transplantation

## Abstract

Juvenile autoimmune hepatitis (JAIH) is severe immune-mediated necro-inflammatory disease of the liver with spontaneous progression to cirrhosis and liver failure if left untreated. The diagnosis is based on the combination of clinical, laboratory and histological findings. Prothrombin ratio is a useful prognostic factor to identify patients who will most likely require a liver transplant by adolescence or early adulthood. JAIH treatment consists of immune suppression and should be started promptly at diagnosis to halt inflammatory liver damage and ultimately prevent fibrosis and progression to end-stage liver disease. The risk of relapse is high especially in the setting of poor treatment compliance. Recent evidence however suggests that treatment discontinuation is possible after a prolonged period of normal aminotransferase activity without the need for liver biopsy prior to withdrawal.

## 1. Introduction

Juvenile autoimmune hepatitis (JAIH) is an immune-mediated necro-inflammatory disease of the liver, of unknown origin, with spontaneous progression to cirrhosis and terminal liver failure, if diagnosis is overlooked and treatment delayed [1]. A complex interaction among genetic susceptibility, exposure to triggering factors and dysregulation of the immune response to antigens expressed on the hepatocyte surface represent the basis of the pathogenesis of the disease.

Specific circulating autoantibodies along with increased concentration of serum IgG and distinctive histopathological aspects identify “classic” AIH or so called “seropositive” AIH [1,2]. Seropositive AIH is divided into two subtypes: AIH type 1 (AIH-1), positive for smooth muscle antibodies (SMA) and/or antinuclear antibodies (ANA); and type 2 (AIH-2), positive for liver kidney microsomal antibody type 1 (anti-LKM-1) and/or anti-liver cytosol type 1 (anti-LC1) [1,2,3,4]. 

JAIH affects all ages and races worldwide with incidence peaks between 10 and 11 years for AIH-1 and between 6 and 7 years for AIH-2. It is characterized by a female predominance, with a ratio of 3:1 for AIH-1 and up to 9:1 for AIH-2 [2,3].

The role of a genetic predisposition has been discussed for decades. To date, it is known that pediatric AIH-1, similarly to that observed in adults, is associated with human leukocyte antigen (HLA) DRB1*03. AIH-2 is associated with specific HLA class II susceptibility alleles; DQB1*0201 is considered the main determinant of susceptibility while DRB1*07/DRB1*03 is associated with the type of autoantibody present. HLA DQB1*0201 is in strong linkage disequilibrium with both HLA DRB1*03 and DRB1*07 [5]. 

## 2. Diagnosis 

The diagnosis of JAIH can be difficult due to the lack of a single specific marker of the disease or gold standard testing. Rather, AIH diagnosis is based on the combination of laboratory (elevated serum IgG, positive autoantibodies) and histological (most commonly interface hepatitis and portal lymphoplasmacytic infiltrate) features in the setting of clinical suspicion and exclusions of other causes of liver disease (e.g., viral hepatitis, drug-induced liver injury, hereditary, metabolic) (Figure 1). Most patients present with non-specific symptoms such as fatigue and abdominal pain; other presentations include an acute viral hepatitis-like onset with nausea, vomiting and anorexia, or an insidious onset, in about 30% of cases, with vague symptoms up to 2 years before diagnosis. On physical examination, hepatomegaly and/or splenomegaly may be present in up to 50% of children at the time of diagnosis [1,2,6]. The clinical and laboratory presentation of JAIH varies from an asymptomatic elevation of serum aminotransferases with increased serum IgG, to acute liver failure, in about 20% of cases, requiring urgent liver transplantation. In several studies, the prevalence of cirrhosis at diagnosis ranges from 17% to 75% of children with JAIH. Acute and severe hepatitis with jaundice even without encephalopathy and with normal INR (international ratio), usually requires hospitalization and close monitoring given the risk of disease progression [7]. The lack of specific diagnostic markers makes the exclusion of other disorders a fundamental aspect of JAIH diagnosis (e.g., viral hepatitis, drug-induced liver damage, Wilson’s disease, hereditary hemochromatosis). Patients with AIH-2 tend to present at a younger median age and have lower rates of cirrhosis at presentation (38% vs. 69%) [8]. In addition, AIH-1 sometimes presents with autoimmune sclerosing cholangitis (ASC)-AIH overlapping syndrome [9].

### 2.1. Serum Autoantibodies

Diagnosis of seropositive JAIH relies on the presence of circulating autoantibodies [10]. Although autoantibodies can also be found in other immune-mediated liver disorders such as Primary Biliary Cholangitis (PBC) and Primary Sclerosing Cholangitis (PSC), their positivity strongly assists the diagnosis of JAIH and allows differentiation into AIH-1 and AIH-2 [1,2,4]. The most common test for autoantibodies screening is indirect immunofluorescence (IFA) which underlies the guidelines from the Committee for Autoimmune Serology of the International Autoimmune Hepatitis Group (IAIHG) drafted in 2004 [11]. This consensus states that IFA on fresh, multi-organ (liver kidney, stomach) rodent slices represents the first-line screening in AIH. In this essay, the diluted serum of the patient is incubated with tissues substrates; the tissue can bind any autoantibody in the serum, targeting antigens present in the substrates, then a washing deletes unbound antibodies and a second, fluorochrome-labelled, anti-human IgG is added, and, after re-washing the substrates are examined by ultraviolet microscopy checking for stains. IFA assay may also be performed on HEp-2 cells (Human Epithelial type 2 cells), derived from human laryngeal carcinoma and completed by antigen-specific techniques such as dot blot or ELISA. Hep2 cells, however, should not be used for screening purposes as they are frequently associated with false positive results, especially in children [10]. The utilization of three rodent tissues helps the detection of all autoantibodies involved in liver diseases [12]. 

ANA is the most non-specific marker of JAIH and gives a nuclear staining on all the three rodent tissues, mostly with a homogeneous pattern, especially in the liver tissue. ANA can also be detected in PBC, PSC, viral hepatitis, non-alcoholic and alcoholic liver diseases, drug-liver injury and non-liver autoimmune diseases such as systemic lupus erythematous, Sjogren’s syndrome and Systemic Sclerosis [13]. The exact mechanism of ANA production and release is not fully known; it seems to be related to hepatocyte damage and loss of B cell tolerance to several nuclear components. In children with AIH, ANA positivity becomes relevant with a titre > 1:160 and it correlates with disease activity [12].

SMA autoantibodies are a group of autoantibodies that react to protein subunits of cytoplasmic filaments (microfilaments, microtubules or intermediate filaments), they can be detected in the arterial walls of kidney, liver and stomach of rodents. On the rodents’ renal substrate, SMA can show three distinct patterns: V (vessels), G (glomeruli) and T (tubules); the VG and VGT patterns are related but not specific of JAIH. The VGT pattern corresponds to “F-actin” reactivity on cultured fibroblast. SMA can be detected in ASC and in rheumatic and infectious diseases. In children, SMA are considered positive at a titer ≥ 1:100 [2,12]. ANA and SMA usually decrease or even disappear over the course of AIH-1 treatment [12]. 

Anti-LKM1 is the serologic marker of AIH-2 [14]; its IFA pattern produces a bright stain on the hepatocyte cytoplasm and on the P3 portion of the proximal renal tubules of rodents [15]. The molecular target of LKM1 is CYP2D6, a member of the hepatic P450 enzyme [16]. Anti-LKM1 may also be present in Hepatitis C virus infection (HCV) and in patients with AIH as part of autoimmune polyendocrinopathy-candidiasis-ectodermal dystrophy (APECED), a monogenic autosomal recessive disorder (AIRE) sharing the same IFA pattern. Anti-LKM1 is considered positive even at low titer [2,10,12,17]. 

Anti-LC1, another AIH-2 marker, can be found as a sole autoantibody or in association with anti-LKM1 [18,19]. Anti-LC1 can be detected by IFA using the standard pattern of murine liver, kidney and stomach producing bright stains on the cytoplasm of liver cells with a sparing of the centrilobular area, however this pattern may be masked by the co-presence of anti-LKM1. The molecular target is the formimino-transferase cyclodeaminase, a highly expressed liver enzyme related to metabolism of folates [20].

Anti-mitochondrial antibody (AMA) is a marker of PBC; rare cases of AMA-positive JAIH have been described, and long-term follow-up of adults with AMA-positive AIH should be warranted for histological signs of PBC [10]. As with ANA and SMA, AMA are absent in AIH-2 [10]. 

Diagnosis of JAIH can be supported by other autoantibodies, less commonly tested but of clinical importance. Anti-soluble liver antigen (anti-SLA) cannot be detected by IFA, but by molecular-based assay. Anti-SLA positivity is usually found in AIH-1 including overlap syndromes but is not found in association with anti-LKM1 or anti-LC1; it is rarely present in hepatitis C and in drug-induced hepatitis [21]. 

Anti-neutrophil cytoplasmic antibody (ANCA) is detected by IFA on fixed human neutrophilic granulocytes with two different staining: cytoplasmic (c-ANCA) and perinuclear (p-ANCA) [10]. As well as in PSC and in inflammatory bowel diseases, p-ANCA can be frequently detected in AIH-1 [12]. To date, IFA remains the gold standard for detecting autoantibodies in the AIH diagnosis despite being a technique requiring time and trained expertise. IFA is complemented by molecular assays based on purified or recombinant antigen. 

### 2.2. Liver Biopsy

The diagnosis of AIH usually requires a liver biopsy with compatible histological findings particularly in non-typical cases. The hallmark of AIH is characterized by a dense inflammatory infiltrate composed of lymphocytes and plasma cells, which cross the limiting plate and invades the surrounding parenchyma. Hepatocytes surrounded by inflammatory cells swell and undergo pycnotic necrosis. Although plasma cells are typically abundant at the interface and within the lobule, their presence in low numbers does not exclude the diagnosis of AIH. When AIH develops acutely or during episodes of relapse, the common histological finding is panlobular hepatitis with bridging necrosis [22]. In a pediatric cohort, in whom the histology of patients with autoimmune liver disease was compared with that of patients with non-autoimmune liver disease, AIH findings included interface hepatitis, portal lymphoplasmacytic infiltrate, rosette formation and emperipolesis. Emperipolesis is the penetration of an intact cell into another intact cell, with both cells maintaining viability. Emperipolesis is present in 65% of AIH patients and hepatocyte rosettes in 33% [23]. Although none of these findings are pathognomonic of AIH, the combination of them is considered highly suggestive of AIH. Histological examination at presentation allows excluding alternative or concomitant diagnoses, classifying the severity of inflammatory activity and assessing the stage of fibrosis. Histological features of AIH presenting with acute liver failure predominate in the centrilobular zone and consist of four main findings: central perivenulitis, plasma cell-enriched inflammatory infiltrate massive hepatic necrosis and lymphoid follicles [24]. 

### 2.3. Non-Invasive Markers of Fibrosis

In numerous studies, more than 14 serum biomarkers of fibrosis have been tested in patients with AIH. From a recent review of the FibroTest, the AST/platelet index (APRI) and the Fibrosis-4 index (FIB-4) were found to be the best non-invasive fibrosis tests in AIH. Gene expression profiles of immune activation, such as IFN-ɣ and IL-22 transcripts, have been recently investigated and are highly expressed in AIH patients compared to healthy controls. There are also numerous cytokines and chemokines that have been associated with immune activation in AIH, including IL-6, IL-8 and IL-21, as well as CCL2, CXCL9-10, however many of these biomarker changes are not related only to AIH but are often seen in other auto/alloimmune disorders [25]. Transient vibration-controlled elastography (VCTE) strongly correlates with the histological stage of fibrosis in AIH. Notably, however, liver stiffness estimated by VCTE is influenced by both fibrosis and inflammation. In fact, in several studies, the use of VCTE at diagnosis has shown a correlation with the histological degree of inflammation rather than the stage of fibrosis. Only after at least 6 months of successful immunosuppressive therapy to reduce liver inflammation can VCTE accurately diagnose cirrhosis and distinguish advanced stages of fibrosis (F3, F4) from less-severe stages (F0–F2) [26]. 

Magnetic resonance elastography (MRE) has also been found to strongly correlate with histological fibrosis in AIH with excellent accuracy (97%), sensitivity (90%), specificity (100%), positive predictive value (100%) and negative predictive value (90%) in advanced liver fibrosis. In addition, MRE assessment of splenic stiffness may have prognostic value for predicting portal hypertension and esophageal varices [27]. Acoustic radiation force imaging (ARFI) assesses liver stiffness by measuring changes in the speed of wave propagation, and the displacements of short-lived bursts of radiated sound waves are interpreted as changes in liver stiffness. ARFI accuracy for cirrhosis exceeds 90% (sensitivity, 93%; specificity, 85%), and meta-analysis results from 13 studies were comparable to VCTE in predicting ≥2 fibrosis stage and cirrhosis. ARFI, however, may overestimate liver fibrosis in patients with massive hepatic necrosis, cholestasis, severe inflammation and liver congestion [28]. 

### 2.4. Diagnostic Scoring Systems

The International Autoimmune Hepatitis Group (IAIHG) diagnostic scoring system was created by an international group of experts in 1993 and simplified in 2008 (Table 1). The IAIHG has devised a diagnostic system for comparative and research purposes, which incorporates several positive and negative scores, including transaminases values, IgG levels, liver-specific autoantibodies, personal history of other autoimmune disorders and characteristic histological features. The sum of these scores suggests whether a diagnosis of AIH is probable or definite. The revised original AIH scoring system has greater sensitivity than the simplified scoring system (100% vs. 95%), while the simplified scoring system has higher specificity (90% vs. 73%) and accuracy (92% vs. 82%), using clinical judgment as the gold standard [3]. In children, a meta-analysis of four studies related to the accuracy of simplified criteria revealed a sensitivity of 77% and a specificity of 95% [29]. The revised original diagnostic scoring system can be applied to children; limitations of the original and simplified scoring systems revised include: 1. Lack of validation by prospective studies; 2. Lack of accuracy in the setting of concomitant PSC, NAFLD/NASH, liver transplant or acute liver failure; 3. Dependence on autoantibody determinations by IFA rather than by enzyme immunoassay [29]. Overall, diagnostic scoring systems can help establish a diagnosis of AIH in difficult cases but are more useful for defining cohorts of patients with AIH for clinical trials. Not to mention that the challenge for any AIH scoring system is to include the minority of AIH patients who have a cholestatic picture, without misdiagnosing PSC patients who do not have an overlap with AIH. 

## 3. Seronegative AIH

Historically, the presence of autoantibodies with elevated serum immunoglobulin in adult patients with active chronic hepatitis (ACH) was able to predict the favorable response to immunosuppression and ultimately lead to the description of AIH [30,31,32]. Soon after it was noted that some patients with autoantibodies-negative ACH or with cryptogenic chronic hepatitis also had a comparable response to immunosuppression [33]. A systematic review of patients successfully treated with immunosuppression led to the full description of the so-called seronegative AIH (sAIH). Patients with sAIH were found to be undistinguishable from those with “seropositive” AIH in terms of demographic, laboratory and histological features as well as in terms of treatment outcomes [34]. Moreover, this group of patients shared several features of immune dysregulation with seropositive-AIH including hypergammaglobulinemia, association with characteristic HLA haplotypes, propensity to relapse and seroreactivity against investigational autoantigens such as auto-antibodies against the liver membrane lipoprotein preparation known as liver-specific membrane lipoprotein (LSP) or against the hepatic asialoglycoprotein receptor (ASGP-R) [35]. Frequency of sAIH varied from 5 to 35% of a reported cohort of AIH being higher in those presenting with acute or fulminant hepatitis [36]. Criteria for diagnosis were limited to the finding of suggestive histological features in patients with acute or chronic hepatitis once viral, toxic or metabolic causes had been excluded [3]. In the 2019 AASLD guidelines, sAIH was finally endorsed and its prevalence was estimated as 20% of AIH [3]. More recently sporadic cases of antibody-negative disease were reported in children and adolescents with concomitant celiac disease [37,38].

In 2016 we reported a series of 38 children with sAIH representing about 17% of 223 children with autoimmune hepatitis (AIH), diagnosed at two paediatric liver centres in Europe, over a 22-year period [36]. In 2021 two further smaller case series were published for a total of 64 paediatric patients described to date [39,40]. Interestingly the proportion of sAIH was similar in all series varying from 17% to 27% even though, in one series, up to 50% of patients developed autoantibodies a few weeks after clinical onset.

The large majority of children with sAIH presented with acute liver failure [36,39]. Criteria for diagnosis of sAIH in children are the same as in adults: negative virological studies, absence of serum AIH-related autoantibodies, exclusion of others causes of liver disease and liver histology compatible with AIH [36]. It is noteworthy that sAIH was not included in the more recent ESPGHAN/NASPGHAN position paper [2] despite the increasing availability of data, mainly in adult patients. 

Seronegative AIH is probably not a unique entity; the first distinction relies on presence or not of hematological abnormalities. Children without hematological disease have, almost in equal proportion, increased or normal immunoglobulin G levels. Seronegative AIH with increased IgG is similar to classical seropositive AIH. Patients often present with symptomatic disease but with signs of chronicity such as splenomegaly and half of them have severe fibrosis or cirrhosis at liver biopsy. In our series, all patients achieved biochemical remission with conventional immunosuppressive treatment, however relapses were frequent as in seropositive-AIH [36]. Withdrawal of therapy was attempted in 50% of children and successful in all. Among patients with hematological comorbidities, defined as the combination of thrombocytopenia, with or without neutropenia and with a normal bone marrow examination, a common (66%) finding was a lymphocyte count < 1000/mm^3^ with a decreased CD4/8 ratio. Anti-platelets or anti-neutrophils antibodies were found in two thirds of cases. In this group of patients, no relapse of hepatitis was observed under or after the withdrawal of therapy when attempted [36]. 

A subgroup of 10 patients with sAIH had associated severe aplastic anemia (SAA) defined as transfusion-dependent pancytopenia. SAA was present at diagnosis in 30% of cases and occurred 1 to 14 months later in the others. Interestingly, these children were older, with median age of 11 years, all presented with acute severe liver disease and most often liver failure. Fibrosis was absent or mild, but centrilobular necrosis, panlobular infiltrate and bridging necrosis were present and often accompanied by a characteristic portal lymphomonocytic infiltrate and interface activity. Lymphocytopenia with low CD4/8 ratio was noted in 80% of cases of SAA-sAIH at diagnosis. In this subgroup of patients, immunosuppression led to remission of sAIH in all but one child who underwent liver transplant. Noteworthy immunosuppression did not prevent the emergence of SAA, which was managed with cyclosporine and anti-lymphocyte globulins. Four of these patients required bone marrow transplant [36]. Ultimately, all patients, except the transplanted one, were able to stop immunosuppression without relapse [36]. 

Hepatitis-associated aplastic anemia is a well-known clinical syndrome with poor prognosis, often leading to end-stage liver disease and refractory bone marrow failure requiring hematopoietic stem cell transplant [41,42,43]. Humoral signs of increased systemic inflammation and immune dysregulation characterize this condition also characterized by low NK cell activity, high-perforin-expressing CD8 T-cells and increased levels of soluble IL-2 receptor as well as sporadic signs of hemophagocytosis at bone marrow examination [42]. It partially overlaps the clinical picture of hemophagocytic lymphohistiocytosis (HLH) even though these patients do not meet criteria for a definite diagnosis of HLH. It seems reasonable that patients with sAIH and SAA described to date represent the milder side of the spectrum of disease as most of them could be successfully cured with immunosuppression [44].

In summary, sAIH is probably a clinical syndrome, which recognizes different mechanisms of disease. We previously proposed stratification in homogeneous groups of patients, which could be the basis for further research [39]. Immunosuppression should be tailored to the specific needs of each subgroup in terms of quality and duration. Larger retrospective series or ideally prospective studies are necessary to establish the best treatment protocol. 

## 4. Management of AIH

Treatment of AIH is based on immune suppression and should be started promptly at diagnosis to halt inflammatory liver damage and ultimately prevent fibrosis and progression to end stage liver disease. Treatment aims to obtain complete biochemical remission defined as strict normalization of serum activities of ALT/AST, and of immunoglobulin G values, negative or very low-titre autoantibodies as well as normalization of the liver function (albumin, clotting factors). The combination of prednisolone or prednisone and azathioprine constitutes the so-called first-line conventional treatment and allows remission to be achieved in most patients. Alternative therapies are used in cases of initial treatment failure or multiple relapses during tapering or discontinuation attempts [4,45] (Table 2).

### 4.1. Conventional Treatment

Prednisone or prednisolone is generally administered orally at 2 mg/kg/day (up to a maximum of 60 mg/day), azathioprine is administered at the initial dose of 1 mg/kg/day, which can be further increased up to 2.5 mg/kg/day until sustained biochemical remission is achieved [4,45,46]. There is no consensus on the best timing of azathioprine introduction. Some centers choose to initiate combination treatment at diagnosis while others advocate waiting 2 weeks before starting azathioprine to assess steroid responsiveness while excluding azathioprine-induced hepatitis [3,47,48]. It seems reasonable that, at least in selected cases, such as those of patients with acute severe AIH and in cirrhotic patients, azathioprine administration is started at incremental doses [48]. Over time, combination therapy is preferred to prednisone monotherapy given azathioprine’s “steroid-sparing” effect, which allows for a more rapid steroid taper, decreasing the risk of prolonged high-dose administration. The goal of this first induction phase of treatment is to induce a rapid and complete clinical and biochemical remission. Over 80% of patients experience a significant biochemical improvement in 6–10 weeks after starting treatment, although the complete normalization of liver enzymes can take several months [49]. Importantly, steroids have been shown to be beneficial also in cirrhotic patients with AIH and liver failure, allowing the recovery of liver function, with avoidance or delay of liver transplant [50]. The risk of severe infections is, however, not negligible in these patients, representing the most frequent cause of morbidity and mortality [50].

Once remission has been achieved, the aim of the treatment is to maintain it at the lowest possible immunosuppressant dose, preventing disease relapse while avoiding the medication’s side effects. Patients are progressively weaned off steroids with a tapering regimen that should be tailored to the individual patient, the severity of their disease and their response to treatment. When steroids are discontinued, patients should be maintained on azathioprine monotherapy for a prolonged period of time prior to attempting the withdrawal of treatment. Unfortunately, the optimal duration of immunosuppressive treatment is unknown and there is insufficient published data to characterize which patients will relapse when off treatment, however the current recommendation is to treat for at least 3 years, attempting the withdrawal of treatment only in cases of persistent and complete biochemical remission with absent or very low titer autoantibodies [2]. Histological proof of remission on the other end has not been shown to be predictive of future relapses and its strict requirement prior to treatment withdrawal is under debate [51,52]. Close monitoring for side effects of prolonged high-dose steroid treatment is mandatory in all patients. Side effects include hyperphagia, weight gain and impaired linear growth, cataracts with visual impairment, osteoporosis and vertebral collapse, hyperglycemia, arterial hypertension, psychosis and serious aesthetic consequences related to the formation of cutaneous striae and acne [1]. Some patients on azathioprine experiences reversible adverse events, the most common being cytopenia. Additional rare side effects include pancreatitis, nausea, vomiting, abdominal pain and other non-specific ones such as myalgias, arthralgias, cutaneous rashes and fever. 

### 4.2. Second- and Third-Line Treatments

Cyclosporine (CSA) is a potent immunosuppressant that was initially developed to prevent organ transplant rejection and has since been used to treat various autoimmune conditions such as uveitis, rheumatoid arthritis and psoriasis. CSA binds with cyclophilin, an intracellular protein, forming a complex that inhibits calcineurin, a necessary enzyme for T-cell activation. This inhibition results in the reduced production of interleukins and subsequent T-lymphocyte activation. CSA has been successfully utilized as a short-term initial bridge therapy or salvage treatment for autoimmune hepatitis in pediatric patients [53]. Additionally, we recently published a study on 20 patients with autoimmune liver disease treated with CSA for a median period of 6.3 years (range: 4–15.5 years). We showed that CSA was effective and safe in the long-term treatment as well suggesting that it could represent a valid alternative to conventional treatment [54]. CSA was administered at a median dose of 5 mg/kg per day with initial target concentration in serum of cyclosporine of 200–250 ng/mL. The most frequent side effects were hypertrichosis and gingival hyperplasia, which were mild and transitory either spontaneously, after dose tapering, or rarely, after CSA withdrawal [54]. More severe side effects such as nephrotoxicity, arterial hypertension and gastrointestinal and neurological toxicity, are not commonly reported in AIH patients but tend to occur in transplant recipients requiring higher doses. 

Mycophenolate Mofetil (MMF), similarly to azathioprine, is a purine antagonist, however it is more powerful and independent from the thiopurine methyltransferase pathway of catabolism. MMF at an initial dose of 20 mg/kg per day has been used in pediatric AIH patients who either did not tolerate azathioprine due to side effects or were refractory to conventional treatment. In a meta-analysis by Zizzo et al., MMF was found to be the second-most-effective drug for conventional treatment-refractory juvenile AIH after cyclosporine (36% vs. 86% remission rate at 6 months) [55]. Although MMF serum levels can be monitored there is a lack of data regarding therapeutic target levels. Side effects of MFM are reported in 3–34% of patients and include headache, diarrhea, dizziness, hair loss and neutropenia. Additionally, MMF is teratogenic, requiring strict counseling in patients of childbearing age [45]. 

Tacrolimus, like cyclosporine, selectively inhibits calcineurin leading to T-cell suppression. It has been used in several adult studies and single-center studies in children with AIH, as first line and in cases of conventional treatment failure or intolerance. Tacrolimus has been administered alone or in combination with steroids, AZA, or MMF, with a wide range of serum trough levels ranging from 1 to 10 ng/mL [45]. In the largest pediatric prospective single-center study, the authors conclude that although Tacrolimus monotherapy is not sufficient to achieve complete remission in all patients, it allows for a significant reduction in the dosage of other medications and their side effects [56]. Tacrolimus is associated with nephrotoxicity, diabetes and neurological and gastrointestinal side effects. 

Budesonide has relatively recently emerged as a promising alternative to prednisone given its fewer systemic side effects due to its hepatic first-pass metabolism. However, a study on pediatric patients showed that budesonide, when used in combination with azathioprine, resulted in a lower proportion of remission compared to prednisone and azathioprine therapy [57]. A recent retrospective study looking at the effectiveness of budesonide versus prednisone as a first-line treatment in adult patients with non-severe acute or chronic autoimmune hepatitis also concluded that the likelihood of achieving a biochemical response was significantly lower in the budesonide group than in the prednisone group; however, budesonide was associated with a significantly lower rate of adverse events [58]. Moreover, budesonide is contraindicated in cirrhotic patients due to an increased risk of side effects.

Rituximab is a chimeric monoclonal antibody that targets the CD20 antigen on B-cells causing B-cell depletion. It has been used successfully in isolated cases of adults and children with AIH. However, in 2019, a multicenter study published on a cohort of 22 adult patients with difficult-to-treat AIH found that rituximab allowed for 71% freedom from AIH flare at 2 years and reduction in steroid dose. While there is limited data available, the successful use of rituximab in treating refractory hepatitis highlights the need for further studies on the role of B-cell-targeting therapies in specific AIH patient [59]. 

### 4.3. Liver Transplant

Indications for liver transplant in AIH are acute liver failure unresponsive to salvage therapy, acute on chronic liver disease, complications of end-stage liver disease and hepatocellular carcinoma [45]. Up to 8% to 16% of children with AIH have been reported to develop end-stage liver disease leading to liver transplantation despite evidence of biochemical remission while on treatment [60,61]. Female teenagers with complications from AIH-1 and children of African American or Latino-American origin are more commonly transplanted when compared to the overall liver transplant population [62]. Patient and graft survivals for children transplanted for AIH in the USA are 95% and 91% at 1 year and 91% and 84% at 5 years, respectively, which is higher than previously reported. Although the risk of lethal infections remains a concern in this patient population subjected to high-dose immunosuppression pre-transplant, the most recent UNOS data did not highlight any deaths secondary to infection in the post-transplant AIH cohort [63]. Notably, about 20% of transplanted patients will develop recurrence of AIH and although all patients should be carefully monitored for it, recurrence should be strongly suspected in case of reappearance of clinical symptoms and signs, increase of transaminases and IgG serum levels, autoantibody positivity, finding of interface hepatitis on liver biopsy and steroid response.

## 5. Long-Term Follow-Up

JAIH is a severe disease with high-risk of relapse due to poor compliance [64,65] and wide variability of reported sustained remission rates after treatment withdrawal ranging from 3 to 87%. [66,67]. In children, the official recommendations prior to attempting treatment withdrawal include normalization of transaminases for 2 to 3 years, normal serum IgG, negative or low titres of serum autoantibodies and no inflammation on liver histology [2]. In adults, on the other hand, liver biopsy is no longer considered a mandatory prerequisite for treatment withdrawal [3]. In addition, there are only limited pediatric reports on the long-term follow-up beyond 10 years of management [67,68].

We recently published data on a long-term observational study of 117 children with AIH, excluding fulminant, seronegative, drug-induced hepatitis and sclerosing cholangitis, who were diagnosed and treated in a single pediatric hepatology center in France [51]. Cirrhosis was present in 80 children at diagnosis. Management consisted of immunosuppression with prednisone with/without azathioprine in most. Attempts at treatment withdrawal under medical supervision were carried out in one of two ways: before 1981, withdrawal was performed after liver histology with a combination of normal aminotransferase activity and serum gamma globulins. Later, having found that liver histology did not reliably predict the lack of relapse after withdrawal [48], another approach was undertaken: when alanine aminotransferase activity remained normal on therapy for at least 1 year, no liver biopsy was performed, and prednisone was progressively decreased over one year, while checking aminotransferases for normalization before each new decrease. Once prednisone was stopped, azathioprine was continued for another 6 months and then stopped if aminotransferases remained normal. A complete biochemical response was defined as normalization of serum alanine aminotransferase and gammaglobulins no later than 6 months after initiation of treatment following the recent recommendations of the IAIHG [69]. Data were available until death and for 8 to 38 years (median, 20 years) after starting treatment in surviving patients. [51].

The results of this study on a large group of children with type 1 or type 2 AIH, with a balanced proportion of type 1 and type 2 patients, a variety of presentation severities at diagnosis and with a 20-year median follow-up in survivors, provide some confirmation of already known information, present data on debated issues and challenge some generally accepted ideas.

### 5.1. No Differences between Type 1 and Type 2 Hepatitis Regarding Outcome

For a long time, it was claimed that AIH type 2 is more severe than type 1. The results of our survey did not show any differences between AIH types 1 and 2, in particular regarding: The proportion of and time to normalization of alanine aminotransferases on immunosuppression.The success of treatment withdrawal.The possibility of sustained normal alanine aminotransferases on treatment with azathioprine only.The overall long-term overall native liver survival.

### 5.2. Major Importance of an Early Diagnosis and Treatment

The results of our study indicate that the presence of cirrhosis, lack of normalization of prothrombin ratio while on treatment, gastroesophageal varices, and gastrointestinal bleeding from varices are related to a longer interval between first symptoms and diagnosis. 

A recent univariate analysis also reported that native liver survival is related to a shorter duration between the onset of first symptoms and time of treatment [70].

Our study provides major arguments to justify the diagnosis and treatment of autoimmune hepatitis without delay after the first signs of liver disease and to search carefully for a liver involvement in children with extra hepatic autoimmune disorders. This is especially significant since cirrhosis can be present in close to half of children seen within 3 months of the first signs of liver disease. Because established cirrhosis was present at diagnosis in the majority of children we report, the long-term outcome of autoimmune hepatitis per se could be further improved by earlier diagnosis and treatment.

### 5.3. Early Prognostic Significance of Normalization of Prothrombin Ratio

Lack of normalization of prothrombin ratio is a major prognostic factor in AIH. In the majority of cases, prothrombin ratio normalizes with therapy and this is possible even with extremely low values [49,69,71], but the persistence of a prothrombin ratio below 70% or its recurrence after an initial normalization was found to be independently associated with a lower probability of overall and native liver survivals [51]. A low prothrombin ratio persisting or recurring during follow-up, even if transaminases normalize, likely indicates that there is no longer room for improvement in the global liver function, and should lead to the consideration of LT before further complications occur.

### 5.4. A Softer Approach to Treatment Withdrawal

Generally JAIH, and even more so AIH type 2, is thought to require lifelong immunosuppression, as relapse is considered inevitable. In our experience, we found a 24% overall relapse-free rate after treatment discontinuation in a population of 117 patients (19% without relapse for ≥4 years) and a 53% relapse-free rate in the subgroup of patients in whom withdrawal was undertaken under medical supervision (42% relapse-free for ≥4 years) [51]. This contrasts with an almost universal relapse rate when patients discontinued treatment on their own. In the majority of patients, the criteria for withdrawal included persistently normal alanine aminotransferase activity but not liver histology. Moreover, the results show no differences in the success or failure of treatment withdrawal under medical supervision at time of puberty compared to at other ages. This is interesting as, to this date, common knowledge was to avoid withdrawal attempts at the time of puberty based on the notion that hormonal changes may increase the risk of immune perturbations [68,72].

In a few cases, treatment discontinuation could be successful also in patients who relapsed after a first treatment withdrawal. 

In previous reports in whom treatment withdrawal was carried out using strict criteria including liver histology, the relapse-free rate ranged from 45% to 87%. Our percentage of relapse-free patients after a first attempt at treatment discontinuation using a softer protocol of withdrawal fits within this range and, except for the results of Deneau et al. [68], it is not significantly different from the ones reported previously. This is not surprising as we previously found that liver histology prior to stopping treatment is not predictive of the success or failure of treatment withdrawal in children [48].

In addition, it is worth noting that the duration of follow-up is longer in our report than in those in the literature and that it is not clear whether some of the previous reports included treatment withdrawal for children with seronegative hepatitis and/or children with drug-induced hepatitis for whom the risk of relapse after treatment withdrawal is lower [36,67,73,74]. Overall, these findings suggest that the stringent criteria recommended before attempting treatment withdrawal could be somewhat mitigated for children, as already suggested for adults [3]. Moreover, in keeping with previous findings in adults [75], we found that a serum ALT activity below 0.5 x ULN at the time of treatment withdrawal may increase the probability of relapse-free survival. The possibility of very late relapses (seen after 20 years of treatment-free follow-up in one patient) and of severe complications of portal hypertension in treatment-free patients, however, requires lifelong monitoring. Whether lifelong low doses of steroid therapy or azathioprine monotherapy are preferable to attempting treatment withdrawal remains open for discussion [76].

## 6. JAIH and Future Pregnancies

JAIH affects a number of young women of childbearing age, some of whom have a long-lasting history of disease and immunosuppression [77]. Moreover, with at least one third of patients having cirrhosis at diagnosis, the population of pregnant women with AIH is very heterogeneous in terms of severity of disease and treatment regimens. There are many aspects of JAIH that may affect pregnancy: presence or not of cirrhosis and related complications; presence of potentially dangerous comorbidities due to concomitant autoimmune diseases such as type 1 diabetes or antiphospholipid syndrome; biochemical remission or active disease at the beginning of pregnancy; ongoing immunosuppression regimen and its potential toxicity or teratogenicity for the fetus. The literature addressing this issue can be roughly divided into two categories: population-based cohort studies [78,79,80] and single-center retrospective reports [81,82,83,84,85]. In population-based studies data, are retrieved from health registries via the ICD code, in one case matched with histopathology report data [78]. The major pitfalls of these studies are the limited amount of clinical information available and possible coding errors. On the other hand, accurate epidemiological analysis could be conducted thanks to the huge size of the study cohort and possibility to build a control group. Single-centers experiences are valuable because diagnosis is well-documented and clinical details are usually available. Limited size and retrospective nature are the limitations. Fertility in AIH depends on disease activity and it is decreased in poorly controlled AIH [83,85] and in cirrhotic patients, 10% of whom are reported to use assisted reproductive technology [83]. During pregnancy, the increase in estrogen and progesterone induces maternal immunity to enter a tolerogenic status to safely host the semi-allograft fetus. Th0 cells are stimulated to differentiate into Treg and Th2 phenotypes that produce high levels of IL-10, IL-4, IL-5 and TGFβ, anti-inflammatory and tolerogenic cytokines [86]. However, after delivery, pregnancy hormones levels abruptly fall and the reactivity of immune system is rapidly restored. Average transaminase levels in AIH patients are significantly lower during gestation, as reported in a recent metanalysis [87]. 

Loss of remission is rare during pregnancy, although it has been reported in as many as 10–15% of cases in the same studies [81,83]. Relapse after delivery, on the other hand, has been reported to occur in up to 50% of cases [83]. More recent studies, however, report lower post-partum relapse rates, which are likely related to the improved knowledge on drugs’ safety and general AIH management during pregnancy. It is known that patients who are in remission before conception have a lower risk of post-partum relapses [87] and that poorly controlled AIH during pregnancy is associated with a greater incidence of adverse outcomes [81]. In terms of maternal outcomes, more recent population studies have not shown an increased rate of caesarean sections in AIH women [78,79]. No difference was found in the incidence of maternal death excluding patients with decompensated cirrhosis [87,88], while almost all studies report an increased risk of gestational diabetes [78,79,80]. Even patients who are off treatment during pregnancy have a higher risk of diabetes [79], suggesting that it is not related to use of steroids but rather to the implicit association of diabetes with other autoimmune diseases. An increased risk of gestational hypertension, preeclampsia, eclampsia and HELLP syndrome was reported in many single-center studies and population-based studies in the US and Sweden [78,79]. It is still debated whether to consider AIH patients as belonging to the high-risk group for other inflammatory diseases and whether low-dose aspirin prophylaxis should be recommended. In terms of fetal outcomes, preterm birth was more common in AIH than in the general population, with a rate of 9–20% [78,79,80]; interestingly, immunosuppression or cirrhosis did not increase this risk. Conversely, the occurrence of a relapse during pregnancy was associated with an increased risk of preterm birth and higher rate of admission to a neonatal intensive care unit [79]. Birth weight < 2500 g but not small-for-gestational-age newborns were associated with AIH in the US and Sweden [78,79] suggesting that low weight is related only to the greater proportion of preterm. No differences in terms of risk of congenital malformations, neonatal mortality and stillbirth were found in any recent population-based studies [78,79,80,87], but only in older ones, which report a rate of fetal loss of up to 27% [83]. The presence of anti-phospholipids and anti-Ro/SSA antigens must be monitored for they carry the risk of adverse outcomes. Only the presence of cirrhosis was also associated with an increased risk of miscarriage in more recent studies [81,84]. There is increasing evidence that remission of disease should be achieved before conception and maintained during pregnancy. A safe pregnancy should be programmed; immunosuppression should not be weaned during gestation with low-dose steroids and azathioprine being safe during pregnancy [85,89,90]. Breastfeeding is also considered safe. A small amount of both drugs can be detected in breast milk but should not have any effect on the infant; however, it is recommended to wait four hours before nursing [91]. 

Although ciclosporin exposure has been associated with hypertension and intrauterine growth retardation and tacrolimus with nephrotoxicity and gestational diabetes when used during pregnancy, calcineurin-inhibitors are now considered safe both during pregnancy and the breastfeeding period. Data are mainly derived from studies on transplanted patients and rheumatologic ones [92]. It must be considered that pregnancy increases the unbound and active form of tacrolimus, resulting in the underestimation of the whole blood level. Consequently, the dosage should not be increased to avoid toxicity. 

Mycophenolate is strongly discouraged due to its known teratogenicity and replacement with azathioprine should be considered [93].

In conclusion, in the majority of women with AIH, pregnancy does not carry substantial risks but remission of disease is strongly recommended. Preconception counseling and multidisciplinary care during gestation and after delivery must be provided.

## 7. Conclusions

JAIH remains to this day a challenging disease to treat due to its severity, progressive nature if untreated and in some patients despite treatment, and its high risk of relapse. Today, overall probabilities of survival in JAIH range from 79 to 98% at 10 years [6,64,94,95] and are 95% and 83% at 20 and 30 years respectively [64]. Native liver survival has been reported as 58%, 85% and 90% at 10 years [6,64,68], 77% and 90% at 20 years [64,68], and 55% at 30 years, respectively [64]. In our experience [51], the overall survival (85%), despite the high incidence of cirrhosis at diagnosis, is somewhat better than in adults [3]. This is likely, at least in part, due the comorbidities present in older patients. The findings of our recent study [51] were welcomed as relevant [94] and indicate that for children and adolescents with JAIH:There are no outcome differences between AIH-1 and AIH-2.Withdrawal of treatment is possible after a prolonged period of normal aminotransferase activity and it may be carried out at the time of puberty. Liver biopsy prior to treatment discontinuation is not always required.Prothrombin ratio is a useful prognostic factor to identify patients who will most likely require LT by adolescence or early adulthood.

## Figures and Tables

**Figure 1 diagnostics-13-02753-f001:**
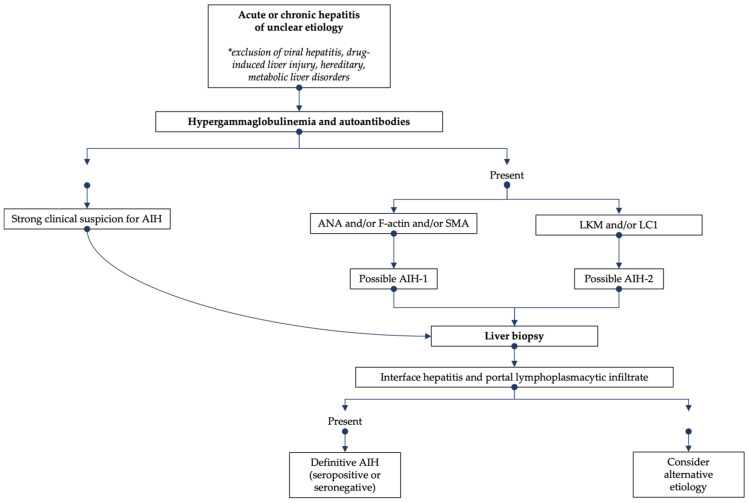
Diagnostic algorithm for the evaluation of suspected AIH.

**Table 1 diagnostics-13-02753-t001:** Simplified diagnostic criteria for autoimmune hepatitis.

Clinical, Laboratory, Histological Features	Points
**Auto antibodies** ANA or SMA ≥ 1:40 ANA or SMA ≥ 1:80 or LKM1 ≥ 1:40 or SLA-positive	+1 +2
**Serum IgG** >upper limit of normal >1.1 times upper limit of normal	+1 +2
**Liver histology *** Compatible with AIH Typical of AIH	+1 +2
**Markers of viral hepatitis** present absent	0 +2
**Aggregate score (pretreatment)**	≥6: probable AIH ≥7: definite AIH

* evidence of hepatitis is a necessary condition.

**Table 2 diagnostics-13-02753-t002:** Treatments for juvenile autoimmune hepatitis: regimens and dosages reported in literature.

Medication	Class of Drug	Regimens	Dose	Side Effects	Additional Information
Prednisone	Corticosteroid	First line	1–2 mg/kg/day	Hyperphagia, weight gain, fat redistribution, cutaneous striae, acne, impaired height growth, severe growth retardation, cataract, osteoporosis, hyperglycemia, arterial hypertension, psychosis	Constitutes the so-called conventional treatment in combination with azathioprine.
Azathioprine	Purine antagonist	First line	1–2.5 mg/kg/day	Cytopenia, pancreatitis, nausea, vomiting, abdominal pain, hepatotoxicity, malignancy	Steroid sparing effect.
Cyclosporine	Calcineurin inhibitor	First line Second line	4–10 mg/kg/day Target trough levels: 100–300 ng/mL	Hypertrichosis, gingival hypertrophy, hypertension, nephrotoxicity	Most effective drug for conventional treatment-refractory JAIH
Mycophenolate mofetil	Purine antagonist	Second line	20–40 mg/kg/day	Leukopenia, headache, diarrhea, dizziness, hair loss	Independent from the thiopurine methyltransferase pathway of catabolism. Teratogenic.
Budesonide	Corticosteroid	First line	6–9 mg/day	Weight gain, fat redistribution, cutaneous striae, acne	Contraindicated in cirrhotic patients.
Tacrolimus	Calcineurin inhibitor	Second line	Target trough levels: 1–10 ng/mL	Bone marrow toxicity, neurotoxicity, nephrotoxicity, opportunistic infections	Unknown optimal through level.
Rituximab	Monoclonal anti CD-20 antibody	Third line	375 mg/m^2^ weekly for 4 weeks, repeated depending on response	Infections, progressive multifocal encephalitis	Only limited data available.

## Data Availability

Not applicable.

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
