# Peer review of "Juvenile Autoimmune Hepatitis: Recent Advances in Diagnosis, Management and Long-Term Outcome"

_diagnostics, 2023, doi:10.3390/diagnostics13172753_

Round 1

Reviewer 1 Report

Dear Editor,

 I would like to congratulate the authors for the excellent review on juvenile autoimmune hepatitis. However, I have a few comments:

R 10:  Author affiliation correction.

R 135: Regarding the performance of the liver biopsy, does it usually represent always or only non-typical cases?

R172: Magnetic resonance elastography (MRE) is the same investigation as iron corrected T1 (cT1), a multiparametric MRI (mpMRI) measure of fibrosis and infammation (fibro-infammation), described by Kamil Janowski et al in reference no. 27?

R675: Reference 27 should be corrected: Janowski K, Shumbayawonda E, Cheng L, Langford C, Dennis A, Kelly M, Pronicki M, Grajkowska W, Wozniak M, Pawliszak P, Chełstowska S, Jurkiewicz E, Banerjee R, Socha P. Quantitative multiparametric MRI as a non-invasive stratification tool in children and adolescents with autoimmune liver disease. Science Rep. 2021 Jul 27;11(1):15261. two: 10.1038/s41598-021-94754-9.

Author Response

R 10:  Author affiliation correction.

The correction was made as suggested

R 135: Regarding the performance of the liver biopsy, does it usually represent always or only non-typical cases?

We thank you the reviewer for his criticism. We modified  the sentence at page 3 (in red) to better clarify that liver biopsy is particularly needed in non-typical cases.

R172: Magnetic resonance elastography (MRE) is the same investigation as iron corrected T1 (cT1), a multiparametric MRI (mpMRI) measure of fibrosis and inflammation (fibro-inflammation), described by Kamil Janowski et al in reference no. 27?

We thank the reviewer for their criticism and we confirm that MRE is the same investigation as MRI described by Janowski now in reference 27

R675: Reference 27 should be corrected: Janowski K, Shumbayawonda E, Cheng L, Langford C, Dennis A, Kelly M, Pronicki M, Grajkowska W, Wozniak M, Pawliszak P, Chełstowska S, Jurkiewicz E, Banerjee R, Socha P. Quantitative multiparametric MRI as a non-invasive stratification tool in children and adolescents with autoimmune liver disease. Science Rep. 2021;11:15261. doi: 10.1038/s41598-021-94754-9.

We apologize for the mistake. The reference 27 was modified as suggested.

Reviewer 2 Report

This article details recent advances in the diagnosis, treatment, and prognosis of autoimmune hepatitis in adolescents in a more complete and accurate manner.

Author Response

This article details recent advances in the diagnosis, treatment, and prognosis of autoimmune hepatitis in adolescents in a more complete and accurate manner

We thank the reviewer 2 for their favorable comments

Reviewer 3 Report

In this review, authors summarize the advances of diagnosis, treatment and long-term outcome of JAIH. This review brings new knowledge on JAIH to readers and the presentation and writing are properly elaborated. The following are some minor issues in this manuscript.

1. It would help to know more about JAIH characteristics. Also, can authors provide a diagnostic algorithm to help readers understand better?

2. What is the gold standard of the JAIH diagnosis?

3. The information of diagnostic scoring systems should be presented more clearly. For instance, a table can be provided.

Author Response

In this review, authors summarize the advances of diagnosis, treatment and long-term outcome of JAIH. This review brings new knowledge on JAIH to readers and the presentation and writing are properly elaborated. The following are some minor issues in this manuscript.

We thank the reviewer for their favorable comments

  1. It would help to know more about JAIH characteristics. Also, can authors provide a diagnostic algorithm to help readers understand better?

The following diagnostic algorithm was added to the text

  1. What is the gold standard of the JAIH diagnosis?

Unfortunately there is no single test or gold standard testing for AIH but rather AIH diagnosis is based on the combination of laboratory (elevated serum IgG, positive autoantibodies) and histological (most commonly interface hepatitis and portal lymphoplasmacytic infiltrate) features in the setting of clinical suspicion and exclusions of other causes of liver disease (e.g. viral hepatitis, drug-induced liver injury, hereditary hemochromatosis, Wilson’s disease).

The following information was added (in red) in the “diagnosis” paragraph.

  1. The information of diagnostic scoring systems should be presented more clearly. For instance, a table can be provided.

The following table with the simplified diagnostic criteria for AIH was added to the text.

Table 1. Simplified Diagnostic Criteria for Autoimmune Hepatitis

Clinical, laboratory, histological features

Points

Auto antibodies

·  ANA or SMA ³ 1:40

·  ANA or SMA ≥ 1:80 or LKM1 ≥ 1:40 or SLA-positive

+1

+2

Serum IgG

·  > upper limit of normal

·  > 1.1 times upper limit of normal

+1

+2

Liver histology*

·  Compatible with AIH

·  Typical of AIH

+1

+2

Markers of viral hepatitis

·  present

·  absent

0

+2

Aggregate score (pretreatment)

≥ 6: probable AIH

≥ 7: definite AIH

*Evidence of hepatitis is a necessary condition